# Efficient photocatalytic hydrogen peroxide generation coupled with selective benzylamine oxidation over defective ZrS₃ nanobelts

Zhangliu Tian[1,2], Cheng Han [1✉], Yao Zhao[2,3], Wenrui Dai[2], Xu Lian [2], Yanan Wang[4], Yue Zheng[4], Yi Shi [2], Xuan Pan[1,4], Zhichao Huang[1,4], Hexing Li [5] & Wei Chen [2,3,4✉]

Photocatalytic hydrogen peroxide (H₂O₂) generation represents a promising approach for artificial photosynthesis. However, the sluggish half-reaction of water oxidation significantly limits the efficiency of H₂O₂ generation. Here, a benzylamine oxidation with more favorable thermodynamics is employed as the half-reaction to couple with H₂O₂ generation in water by using defective zirconium trisulfide (ZrS₃) nanobelts as a photocatalyst. The ZrS₃ nanobelts with disulfide ($S_2^{2-}$) and sulfide anion ($S^{2-}$) vacancies exhibit an excellent photocatalytic performance for H₂O₂ generation and simultaneous oxidation of benzylamine to benzonitrile with a high selectivity of >99%. More importantly, the $S_2^{2-}$ and $S^{2-}$ vacancies can be separately introduced into ZrS₃ nanobelts in a controlled manner. The $S_2^{2-}$ vacancies are further revealed to facilitate the separation of photogenerated charge carriers. The $S^{2-}$ vacancies can significantly improve the electron conduction, hole extraction, and kinetics of benzylamine oxidation. As a result, the use of defective ZrS₃ nanobelts yields a high production rate of 78.1 ± 1.5 and 32.0 ± 1.2 μmol h⁻¹ for H₂O₂ and benzonitrile, respectively, under a simulated sunlight irradiation.

[1] SZU-NUS Collaborative Innovation Center for Optoelectronic Science & Technology, International Collaborative Laboratory of 2D Materials for Optoelectronics Science and Technology of Ministry of Education, Institute of Microscale Optoelectronics, Shenzhen University, Shenzhen, China. [2] Department of Chemistry, National University of Singapore, 3 Science Drive 3, Singapore, Singapore. [3] Joint School of National University of Singapore and Tianjin University, International Campus of Tianjin University, Binhai New City, Fuzhou, China. [4] Department of Physics, National University of Singapore, 2 Science Drive 3, Singapore, Singapore. [5] International Joint Lab on Resource Chemistry, College of Chemistry and Materials Science, Shanghai Normal University, Shanghai, China. ✉email: hancheng@szu.edu.cn; phycw@nus.edu.sg

Artificial photosynthesis, i. e. the conversion of solar energy into chemical energy, is considered as one of the promising approaches to synthesize chemicals with the unlimited energy source and minimized environmental problems[1,2]. As a promising liquid solar fuel generated by artificial photosynthesis, hydrogen peroxide ($H_2O_2$) has attracted growing attention because of its high commercial value and low transportation cost[3]. Substantial efforts have been devoted to the development of effective photocatalysts or photocathodes for the $H_2O_2$ generation from water and $O_2$[4–6]. In the photocatalytic process, the sluggish oxidation of water induced by the photogenerated valence holes is a limiting factor for the production of $H_2O_2$[7–11]. Most previous reports focused on improving the half-reaction of $O_2$ reduction, e. g. by consuming holes with sacrificial agents, such as isopropyl alcohol, benzyl alcohol, and 2-PrOH[3,12,13]. However, the development of an alternative oxidation reaction with accelerated kinetics to produce value-added chemicals was rarely reported. On the other hand, selective oxidation of amines to nitriles with lower oxidation potential than water plays an vital role in both laboratorial and industrial synthetic process since nitriles are the important intermediates during the synthesis of fine chemicals, pharmaceuticals, and agrochemicals[14–21]. Intensive research has been carried out to synthesize nitriles from primary amines through dehydrogenation[22–28]. However, most of the reactions are conducted in organic solvents under harsh conditions, such as high-temperature, exposure to high-pressure oxygen or air, and presence of oxidants. Photocatalytic reactions have been demonstrated to be an effective approach to synthesize nitriles under mild conditions[29–31], but previous works were seriously limited by the use of noble metals as the co-catalysts to realize the dehydrogenation in organic solvents. Thus, the development of artificial photosynthesis in an aqueous and easy scale-up condition with earth-abundant photocatalysts is highly desirable for the production of nitriles in an economically-viable and environment-friendly way.

Monoclinic zirconium trisulfide ($ZrS_3$) (ICCD PDF no. 30-1498), a layered n-type transition metal trichalcogenide (TMT), has recently drawn great research interest due to the extraordinary properties arising from its unique disulfide anions ($S_2^{2-}$)[32]. $ZrS_3$ has shown a good optical responsivity of 290 mA $W^{-1}$ with an in-plane hole and electron mobility at a magnitude of $10^2$ and $10^3$ $cm^2$ $V^{-1}$ $s^{-1}$, respectively[33–36]. In particular, $ZrS_3$ possesses a bandgap of ~2 eV with a more negative conduction band minimum (CBM) than the $H_2$ evolution potential[37], making $ZrS_3$ a promising semiconductor for photocatalytic and photoelectrochemical applications. The previous studies on zirconium nitride have demonstrated that its superior performance for $O_2$ reduction stems from the interaction between Zr sites and oxide species, where the Zr d-orbitals make a strong contribution[38]. Intriguingly, the conduction band of $ZrS_3$ is mainly composed of Zr d-orbitals[33,35], which has a much more negative potential than the reducing potential of $O_2$ to $H_2O_2$. Therefore, $ZrS_3$ shows great potential for the photocatalytic $H_2O_2$ generation.

There are three categories of S ($S_1$, $S_2$, and $S_3$) environments in monoclinic $ZrS_3$ lattice, where $S_1$ denotes the sulfide ion ($S^{2-}$) and $S_2$, $S_3$ are interpreted as the $S_2^{2-}$ (Fig. 1a, b)[39]. Recently, both theoretical calculations and experimental investigations have demonstrated that moderate $S_2^{2-}$ vacancies can greatly promote the separation of photogenerated charge carriers in TMTs (Supplementary Fig. 1a and b)[40,41]. In addition, the anion vacancies existing on the surface of n-type semiconductors can further improve its photocatalytic and photoelectrochemical performance by accelerating the kinetics of hole transfer on the surface[42–45]. The crystal structure analysis of $ZrS_3$ inspires us that the $S_2^{2-}$ and $S^{2-}$ vacancies can be separately introduced into $ZrS_3$ by different methods (Fig. 1 and Supplementary Fig. 1). Experimentally, hexagonal $ZrS_2$ (ICCD PDF no. 11-0679) is usually obtained by

vacuum annealing of monoclinic $ZrS_3$ at elevated temperatures. This suggests that $ZrS_3$ can desulfurize into $ZrS_2$ by the post-annealing at a higher temperature under vacuum[46]. Only one type of Zr ($Zr_1$) and S ($S_1$) environment exists in $ZrS_2$, where the $Zr_1$-$S_1$ bond length is similar to that in $ZrS_3$ (Fig. 1a–c). Besides, $ZrS_2$ shows a similar layered structure to $ZrS_3$, and atomic layers in both materials are parallel to the (001) plane (Supplementary Fig. 1c–f). When $ZrS_3$ transforms into $ZrS_2$, it does not need much tweaking of the framework along both [010] and [001] directions (Supplementary Fig. 1c and e), but it is required to adjust the framework along the [100] direction (Fig. 1d and f). Figure 1 clearly suggests that the $ZrS_3$ can transform into $ZrS_2$ by two steps: the first step can be the desulfurization of $ZrS_3$ to release $S_2$ or $S_3$ ions to form a distorted crystal structure of $ZrS_2$ (Fig. 1a, b, d, and e), and then the distorted crystal structure undergoes structural relaxation by tuning the length and angle of Zr-S bonds to form $ZrS_2$ without breaking or regrouping the bonds (Fig. 1b, c, e, and f). Thus, the high-temperature vacuum annealing is expected to be an effective scheme to produce $S_2^{2-}$ vacancies in $ZrS_3$. On the other hand, $S^{2-}$ ions have high adaptability when coordinated with metal ions, which can serve as either terminal or bridge ions to interact with metals (especially for alkali metals). This different from the $S_x^{2-}$ ($x \geq 2$) ions that are difficult to bond with metals (Supplementary Fig. 1g)[47]. Moreover, $ZrS_3$ is easily formed as nanobelts (NBs) with rich $S^{2-}$ ions exposed at the edges (Supplementary Fig. 1h and i). Previous studies have used active metals such as Mg, Al, and Zn to induce oxygen vacancies in metal oxide due to their reducibility[48]. Compared to these metals, alkali metal lithium (Li) has a higher reducibility and can be easily intercalated into host materials. Li can be easily dissolved in solvents like ammonia or ethanediamine to form Li-based complex for the solvothermal treatment, which has been widely utilized to enhance the transition temperature of superconducting materials[49–51] and in particular to induce oxygen vacancies on $TiO_2$[52]. Therefore, such Li-based treatment could be an effective approach to induce $S^{2-}$ vacancies on $ZrS_3$.

Here, $ZrS_3$ NBs with both $S_2^{2-}$ and $S^{2-}$ vacancies are employed to enhance the photocatalytic production of $H_2O_2$ coupled with the selective oxidation of benzylamine to benzonitrile in water. The impacts of $S_2^{2-}$ and $S^{2-}$ vacancies on modulating the charge carrier dynamics and photocatalytic performance are systematically investigated. The $S_2^{2-}$ vacancies can significantly facilitate the separation of photogenerated charge carriers; while the $S^{2-}$ vacancies are demonstrated to not only promote the electron conduction and hole extraction in the photocatalytic process but also improve the kinetics of benzylamine oxidation. As a result, the use of defective $ZrS_3$ NBs as photocatalyst produces $H_2O_2$ and benzonitrile at a high rate of 78.1 ± 1.5 and 32.0 ± 1.2 μmol $h^{-1}$ respectively, under the illumination of a simulated sunlight.

## Results and discussions

**Structural properties and band structures of photocatalysts.** $ZrS_3$ NBs were synthesized via a chemical vapor transport of S powder to Zr powder using iodine as a transport agent. $ZrS_3$ with $S_2^{2-}$ vacancies ($ZrSS_{2-x}$) was obtained by the re-annealing of the as-grown $ZrS_3$ NBs at 700 °C for different time (10, 15, and 20 min) under vacuum. $ZrSS_{2-x}$ with $S^{2-}$ vacancies ($ZrS_{1-y}S_{2-x}$) was prepared through a low-temperature solvothermal treatment by using Li-dissolved ethanediamine with different amounts of Li (50, 100, and 150 mg). We denote $ZrSS_{2-x}$ NBs annealed for X time as $ZrSS_{2-x}$(X) and $ZrS_{1-y}S_{2-x}$ NBs annealed for X min and treated with Y mg Li as $ZrS_{1-y}S_{2-x}$(X/Y). The x-ray diffraction (XRD) pattern indicates the formation of $ZrS_3$ in the monoclinic phase (ICCD PDF no. 30-1498), and the vacuum annealing and

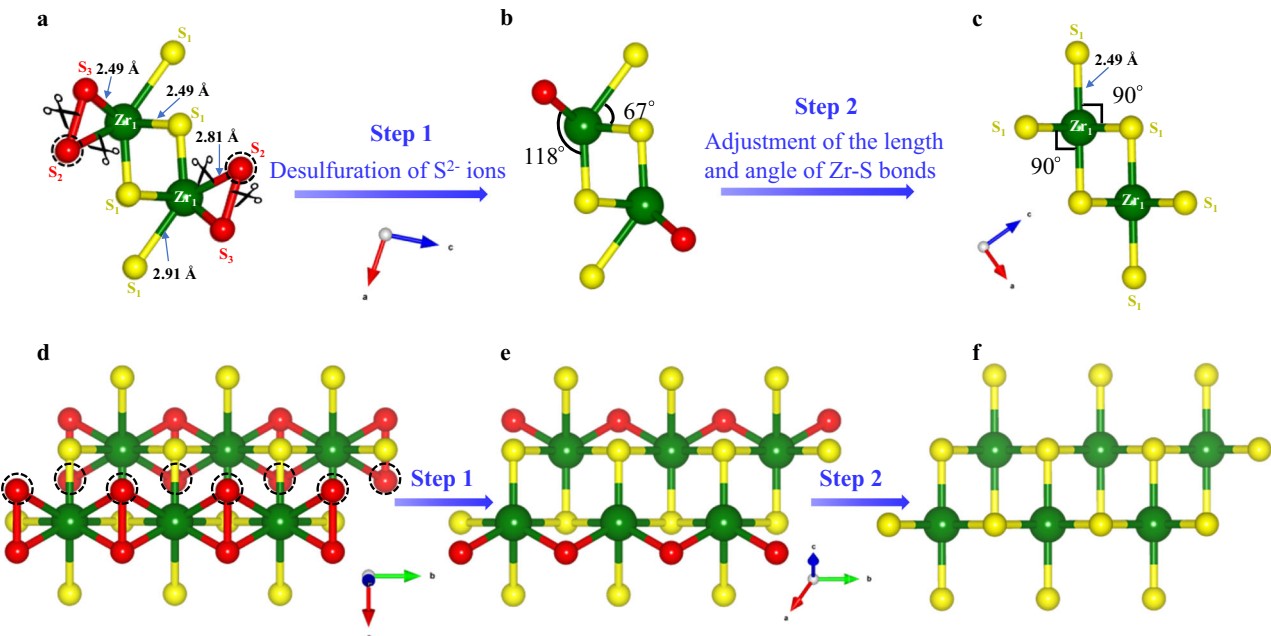

**Fig. 1 The transformation of the crystal structure of ZrS₃ into ZrS₂.** The schematic process of the transformation of monoclinic ZrS₃ (ICCD PDF no. 30-1498) into hexagonal ZrS₂ (ICCD PDF no. 11-0679) from the [010] (**a–c**) and [001] (**d–f**) views. **a**, **d** Crystal structure of monolayer ZrS₃ with a boundary of 1 x 3 x 1 from the [010] and [001] views, respectively. **b**, **e** Crystal structure of monolayer ZrS₃ after desulfuration of $S_2^{2-}$ ions from **a** and **d**, respectively. **c**, **f** Crystal structure of monolayer ZrS₂ with a boundary of 1 x 3 x 1 from the [010] and [001] views, respectively.

further Li treatment did not induce any phase transition in ZrSS$_{2-x}$(15) and ZrS$_{1-y}$S$_{2-x}$(15/100) NBs (Supplementary Fig. 2). However, the samples show decreased peak intensity from the ZrS₃ to the ZrS$_{1-y}$S$_{2-x}$(15/100) NBs, due to the introduction of sulfur vacancies that reduce the crystallinity of ZrS₃, as observed in the high-resolution transmission electron microscopy (HRTEM) image (Supplementary Fig. 3). The obtained ZrS₃ was formed as NBs with the width ranging from 300 nm to 3 μm and length in tens of micrometers (Fig. 2a–c). Scanning electron microscope (SEM) and atomic force microscope (AFM) measurements were further conducted to statistically determine the length, width, and thickness distribution of ZrS₃ NBs. The average length, width, and thickness of NBs were measured to be 24 μm, 840 nm, and 38 nm, respectively (Supplementary Figs. 4 and 5), where all the histograms exhibit a unimodal distribution with the peak in the range of 20–30 μm, 0.6–1.0 μm and 25–45 nm, respectively (Supplementary Fig. 5d). As a result, the average ratio of width/thickness was calculated to be ~22, which qualifies the label of "NBs" for our samples. The individual ZrS₃ NB is confirmed as the single crystal along [010] direction by the TEM and corresponding selected area electron diffraction (SAED) characterization (Fig. 2d). It is demonstrated that the ZrS₃ layer is parallel to the axial direction of NB, which is in favor of charge carrier transport[40]. As shown in the HRTEM images (Supplementary Fig. 3), ZrS₃ exhibits highly-ordered lattice fringes with an excellent crystallinity, while ZrS$_{1-y}$S$_{2-x}$(15/100) shows an obvious lattice disorder with relatively poor crystallinity. This suggests that the introduction of sulfur vacancies could lead to a decreased crystallinity in ZrS₃, consistent with the XRD results.

As shown in the diffuse reflectance UV–vis spectra (Fig. 2e), both ZrS₃ and ZrSS$_{2-x}$ (15) NBs absorb light with the wavelength up to ~650 nm, corresponding to a bandgap of 2.02 eV (Supplementary Fig. 6). ZrS$_{1-y}$S$_{2-x}$(15/100) NBs present a slight red-shift of absorption spectrum, revealing a smaller bandgap of 1.98 eV. The Mott−Schottky plots for all three samples exhibit positive slopes, indicating the n-type behavior of ZrS₃ (Fig. 2f).

These results were obtained by measuring the photocatalysts deposited on the fluorine-doped tin oxide (FTO) substrate. It is worth noting that the deposition process did not induce any obvious change of the photocatalyst (Supplementary Fig. 7a–c), suggesting that the sample on the FTO substrate measured with the photoelectrochemical set-up is essentially the same photocatalyst. The flat band potentials ($E_{fb}$) of ZrS₃, ZrSS$_{2-x}$(15), and ZrS$_{1-y}$S$_{2-x}$(15/100) are estimated to be −0.10, −0.11, and −0.18 V versus reversible hydrogen electrode ($V_{RHE}$), respectively (Fig. 2g and Supplementary Fig. 7d). $E_{fb}$ is commonly used to estimate the CBM for a series of n-type semiconductors at the surface in an aqueous environment, which agreed with their theoretically determined values[1,44,53–55]. Previous studies on the energy positions of semiconductors have shown that the CBM of zirconium-based sulfides is very close to their $E_{fb}$[37,54], and therefore the CBM of ZrS₃, ZrSS$_{2-x}$(15), and ZrS$_{1-y}$S$_{2-x}$(15/100) can be directly determined by their $E_{fb}$. Based on the Mott −Schottky (Fig. 2f, g, and Supplementary Fig. 7d) and UV–vis spectra results, the CBM and valance band maximum (VBM) for ZrS₃, ZrSS$_{2-x}$(15), and ZrS$_{1-y}$S$_{2-x}$(15/100) were revealed to be −0.10, −0.11, −0.18 $V_{RHE}$ (CBM) and 1.92, 1.91, and 1.80 $V_{RHE}$ (VBM), respectively (Fig. 2h and Supplementary Fig. 8). The CBMs of ZrS₃, ZrSS$_{2-x}$(15), and ZrS$_{1-y}$S$_{2-x}$(15/100) are higher than the potential for two-electron reduction of O₂. Previous studies have shown that the oxidation potential of benzylamine lies higher than that of water, and the benzylamine oxidation was thus used to replace oxygen evolution reaction to couple with photocatalytic and electrocatalytic hydrogen evolution reaction[14,17,55,56]. This suggests the VBM of defective ZrS₃ NBs lying far below the oxidation potential of benzylamine, indicating that these photocatalysts are applicable to the photocatalytic O₂ reduction and benzylamine oxidation.

**Characterizations of vacancy structure.** Four characteristic Raman modes of ZrS₃ located at ~ 147, 274, 315, and 524 cm$^{-1}$ were observed in Fig. 3a, which are assigned to the rigid chain vibration (I: A$_g^{rigid}$), internal out-of-plane vibrations

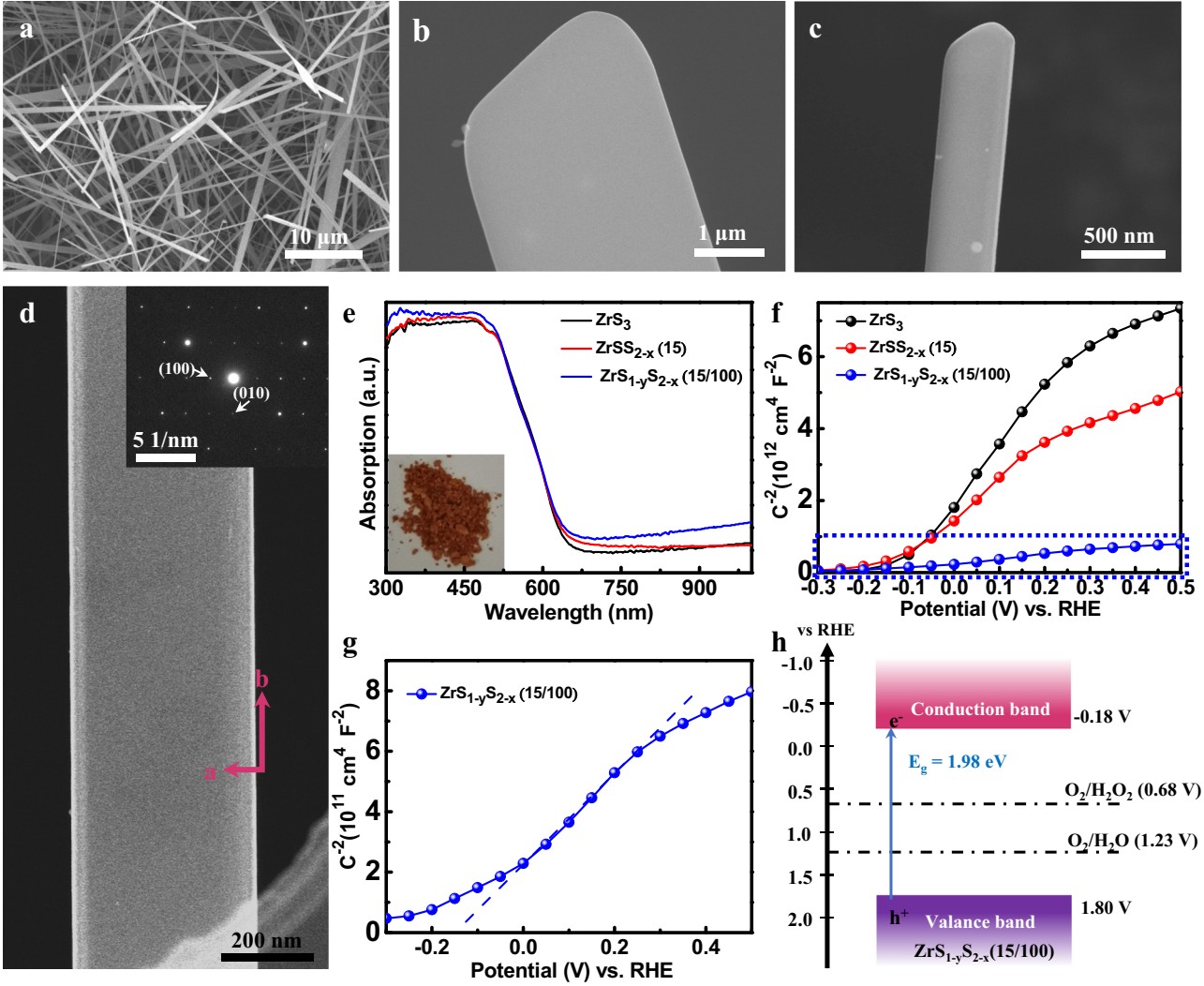

**Fig. 2 Structural properties and band structures of defective ZrS₃ NBs. a** Top-sectional, **b**, **c** high-magnification SEM images of the ZrS$_3$ NBs. **d** TEM image and SAED pattern of single ZrS$_3$ NB. **e** Diffuse reflectance UV-vis spectra of the ZrS$_3$, ZrSS$_{2-x}$(15) and ZrS$_{1-y}$S$_{2-x}$(15/100) NBs. Inset, the photograph of the ZrS$_{1-y}$S$_{2-x}$(15/100) NBs. **f** Mott–Schottky plots of ZrS$_3$, ZrSS$_{2-x}$(15) and ZrS$_{1-y}$S$_{2-x}$(15/100) NBs and **g** Mott-Schottky plot of ZrS$_{1-y}$S$_{2-x}$(15/100) magnified from **f**. **h** Schematic band structure diagram for ZrS$_{1-y}$S$_{2-x}$(15/100). To have a clear view of the single NB, the sample for this SEM measurement was prepared by evaporating the isopropanol dispersion of ZrS$_3$ NBs.

(II: $A_g^{internal}$ and III: $A_g^{internal}$), and S–S diatomic motion (IV: $A_g^{s-s}$), respectively[35]. The Raman spectra show an obvious red-shift of $A_g^{s-s}$ mode by ~5 cm$^{-1}$ from ZrS$_3$, ZrSS$_{2-x}$(15), and ZrS$_{1-y}$S$_{2-x}$(15/100), originating from the introduction of S$_2^{2-}$ vacancies[40]. We also observed a ~3 cm$^{-1}$ red-shift of $A_g^{rigid}$ mode from ZrS$_3$ and ZrSS$_{2-x}$(15) to ZrS$_{1-y}$S$_{2-x}$(15/100). Since the $A_g^{rigid}$ is correlated to the vibration of quasi-one-dimensional chains in the direction of c axis (Supplementary Fig. 9a), the shift of $A_g^{rigid}$ mode in ZrS$_{1-y}$S$_{2-x}$(15/100) results from the introduction of S$^{2-}$ vacancies, which alters the length of Zr–S bonds within each chain. The similar shift of $A_g^{rigid}$ mode was also identified from ZrS$_3$ and ZrSS$_{2-x}$(15) to the only Li-treated ZrS$_3$ with 100 mg Li (ZrS$_{1-y}$S(100)) as shown in Supplementary Fig. 9b. To explore the effect of process parameters during the synthesis of defective materials on the types and density of defects, and the correlation with photocatalytic activity, orthogonal experiments have been performed by simultaneously changing the Li amount and vacuum annealing duration. All the samples were further examined by the Raman characterization, and Supplementary Fig. 10a–d show representative Raman spectra of the defective ZrS$_3$ NBs separately

treated by the Li-treatment with different Li amount and by the vacuum annealing for different time. The gradual red-shift of $A_g^{rigid}$ mode (difference from 1.6 to 6.1 cm$^{-1}$) was observed with increasing Li amount, resulting from the increased concentration of S$^{2-}$ vacancies in Li-treated ZrS$_3$ NBs (Supplementary Fig. 10a and b). Similarly, the vacuum annealing triggered a red-shift of $A_g^{s-s}$ mode due to the generated S$_2^{2-}$ vacancies[40], which was enlarged from 2.4 to 8.1 cm$^{-1}$ by prolonging the annealing time (Supplementary Fig. 10c and d). Based on the Raman results, the shifts of both $A_g^{rigid}$ and $A_g^{s-s}$ modes for all the samples were extracted and plotted as 3D histograms, as shown in Supplementary Figure 11a and b. The $A_g^{rigid}$ shift only depends on the Li amount, while the $A_g^{s-s}$ shift only relies on the annealing time. These results reveal that the S$_2^{2-}$ and S$^{2-}$ vacancies in ZrS$_{1-y}$S$_{2-x}$ can be independently induced by the vacuum annealing and Li-treatment respectively, and the concentration of vacancies can be further controlled by varying the annealing time and Li amount.

The XPS characterization was conducted on these samples to further confirm the vacancy type. After the vacuum annealing, the ZrSS$_{2-x}$(15) NBs exhibit a slightly lower binding energy of the

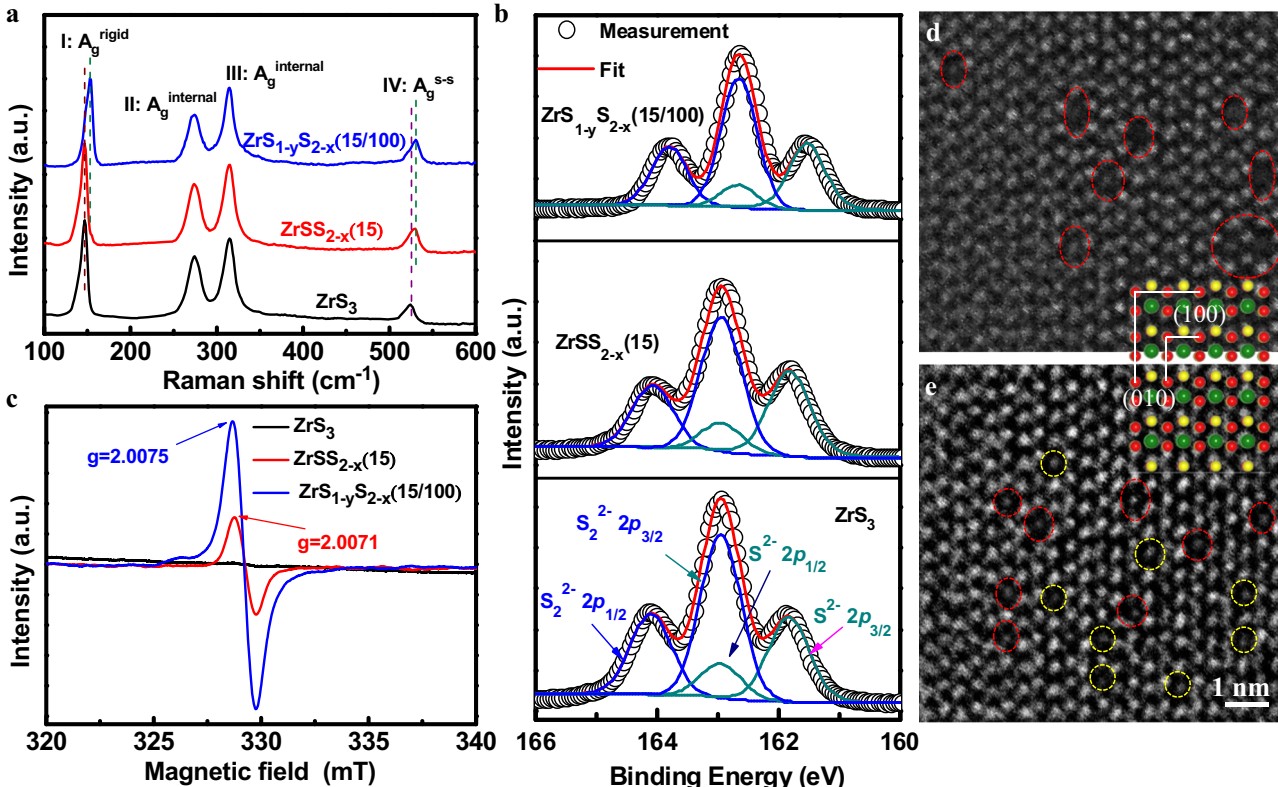

**Fig. 3 Characterizations of vacancy structure of defective ZrS₃ NBs. a** Raman spectra, **b** S 2p XPS spectra, and **c** EPR spectra of the ZrS₃, ZrSS₂₋ₓ(15) and ZrS₁₋ᵧS₂₋ₓ(15/100) NBs. HAADF-STEM images of **d** ZrSS₂₋ₓ(15) and **e** ZrS₁₋ᵧS₂₋ₓ(15/100) measured from a spherical aberration-corrected TEM. Inset: the crystal lattice of ZrS₃ along the [001] orientation. The red and yellow circles represent $S_2^{2-}$ and $S^{2-}$, respectively.

Zr 3$d$ core level than ZrS₃ NBs, consistent with the results from ZrS₃ to ZrS₂ (Supplementary Fig. 12a)[57]. Furthermore, ZrSS₂₋ₓ(15) shows a significant attenuation of $S_2^{2-}$ 2p peaks with the nearly unchanged $S^{2-}$ 2p peaks compared to ZrS₃ (Fig. 3b and Supplementary Fig. 12b), indicating the mere increase of $S_2^{2-}$ vacancies in ZrSS₂₋ₓ(15). It is worth noting that no clear peak shift was observed for ZrSS₂₋ₓ(15), which results from the almost retained electron density around the S sites, as revealed by the Mott-Schotty results. After further Li treatment, both Zr 3$d$ and S 2p core levels of ZrS₁₋ᵧS₂₋ₓ(15/100) NBs shifted to the lower binding energy by ~0.3 eV regarding ZrSS₂₋ₓ(15) (Supplementary Fig. 12a, c, and d), due to the increased electron density around the S sites induced by $S^{2-}$ vacancies[58]. In particular, the intensity of $S^{2-}$ 2p peaks in the ZrS₁₋ᵧS₂₋ₓ(15/100) was clearly lower than that of ZrSS₂₋ₓ(15) (Fig. 3b and Supplementary Fig. 12d), revealing the increase of $S^{2-}$ vacancies. This phenomenon has been commonly observed in the transition metal sulfide with $S^{2-}$ vacancies such as MoS₂, In₂S₃, and CuInS₂[59–61]. The similar variation of Zr 3$d$ and S 2p spectra observed from ZrS₃ to ZrS₁₋ᵧS (100) further suggest the separate introduction of $S^{2-}$ vacancies by the Li treatment (Supplementary Fig. 12a and c). The type and density of sulfur vacancies for all the samples were further quantitatively analyzed by the XPS characterization, and representative S 2p XPS spectra of the defective ZrS₃ NBs separately treated by the vacuum annealing and Li-treatment are presented in Supplementary Fig. 13a and b, respectively. The x and y values in the label of ZrS₁₋ᵧS₂₋ₓ for all the samples were estimated from the XPS results by calculating the area ratio of characteristic peaks in defective samples to that of ZrS₃, as summarized in Supplementary Table 1. Agreed with the Raman results, the vacuum annealing and Li-treatment can independently attenuate the intensity of $S_2^{2-}$ 2p and $S^{2-}$ 2p peaks,

respectively, as revealed by the almost unchanged x and y values under the identical annealing time and Li amount, respectively. For an intuitive comparison, the x and y values as a function of the annealing time and Li amount were plotted in 3D histograms, as shown in Supplementary Fig. 14a and b, respectively. The x was estimated to be 0.20 ± 0.01, 0.36 ± 0.01, and 0.49 ± 0.01 for the annealing time of 10, 15, and 20 min, respectively; while the y was evaluated to be 0.05 ± 0.01, 0.36 ± 0.01, and 0.49 ± 0.01 for 50, 100, and 150 mg Li, respectively.

In addition, the electron paramagnetic resonances (EPR) investigation was also carried out to detect the vacancy structure. A characteristic peak can be clearly detected at g = 2.0071 for all the vacuum annealed samples in Supplementary Fig. 15a, and the peak intensity is proportional to the vacuum annealing duration. This suggested the characteristic peak of Zr-$S_2^{2-}$ dangling bonds is located at g = 2.0071[58]. Similarly, the characteristic peak of Zr-$S^{2-}$ dangling bonds is located at g = 2.0082 (Supplementary Fig. 15b). Therefore, the characteristic peak located at g = 2.0075 for ZrS₁₋ᵧS₂₋ₓ(15/100) NBs suggests the formation of both $S^{2-}$ and $S_2^{2-}$ vacancies (Fig. 3c). The higher signal intensity of ZrS₁₋ᵧS₂₋ₓ(15/100) than that of ZrSS₂₋ₓ(15) indicates more sulfur vacancies existing in ZrS₁₋ᵧS₂₋ₓ(15/100) NBs[60]. To have a direct view of the atomic arrangement for ZrSS₂₋ₓ(15) and ZrS₁₋ᵧS₂₋ₓ(15/100) NBs, the high-angle annular dark-field scanning transmission electron microscopy (HAADF-STEM) images were obtained, where the atomic sites can be determined by comparing the HAADF-STEM image with the crystal structure of ZrS₃ lattice along the [001] direction (Supplementary Fig. 15c and d). The ZrSS₂₋ₓ(15) demonstrates the missing atoms only emerging on the $S_2^{2-}$ sites, as indicated by the red dashed circles in Fig. 3d, while the atomic vacancies exist on both $S^{2-}$ (indicated by yellow dashed circles) and $S_2^{2-}$ sites for ZrS₁₋ᵧS₂₋ₓ(15/100) (Fig. 3e).

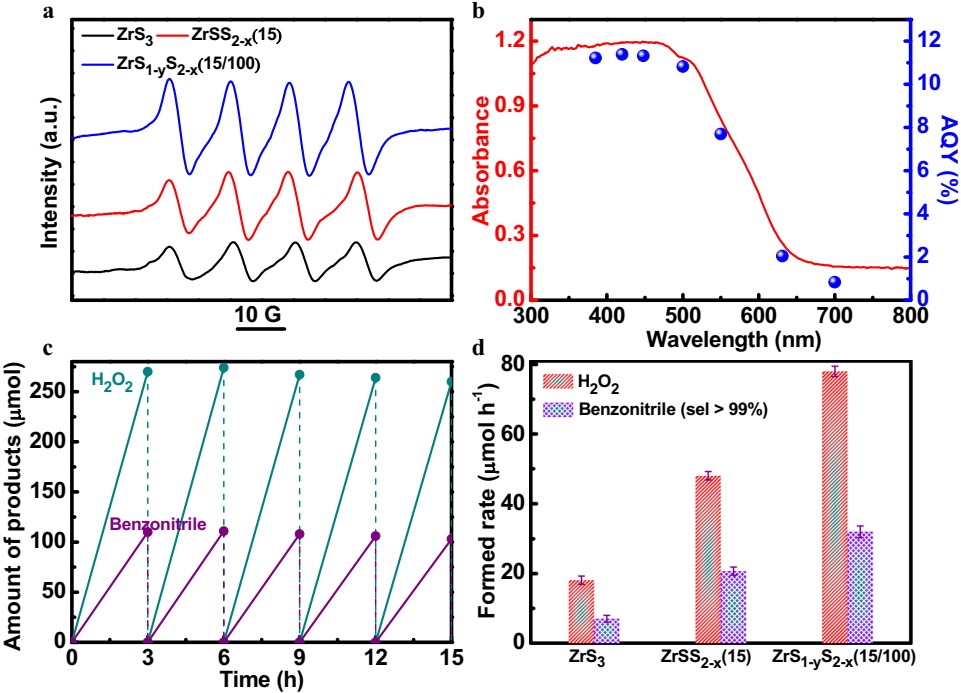

**Fig. 4 The photocatalytic properties of defective ZrS₃ NBs. a** EPR spectra of ZrS₃, ZrSS₂₋ₓ(15) and ZrS₁₋ᵧS₂₋ₓ(15/100) in the presence of DMPO. **b** Absorption spectrum of ZrS₁₋ᵧS₂₋ₓ(15/100) and its dependence of AQY with monochromatic light irradiation. Conditions: 30 ml aqueous solution with 1 mmol benzyl alcohol, 50 mg photocatalysts. **c** Results of H₂O₂ and benzonitrile generation for a repeated photoreaction sequence with ZrS₁₋ᵧS₂₋ₓ(15/100) under AM1.5G simulated sunlight irradiation. **d** H₂O₂ and benzonitrile evolution rate by the respective photocatalysts under AM1.5G simulated sunlight irradiation. Error bars are the standard error of the mean for 9 independent samples. Conditions: 30 ml H₂O with 1 mmol benzylamine, 50 mg photocatalysts, 1 bar O₂.

**Photocatalytic performance.** The photocatalytic capability of the defective ZrS₃ NBs for reducing O₂ to create the reactive oxygen species (ROS) was first evaluated by the EPR trapping experiment using 5,5-dimethyl-1-pyrroline N-oxide (DMPO). As illustrated in Fig. 4a, four characteristic peaks of DMPO−O₂•⁻ were observed for all NBs, confirming the generation of O₂•⁻[60,62,63]. The introduction of $S_2^{2-}$ vacancies was found to enhance the reduction of O₂ to O₂•⁻, and the additional introduction of $S^{2-}$ vacancies led to a further increased photocatalytic activity. The correlation of types and density of sulfur vacancies with photocatalytic activity was further examined by the iodometry[12] under the irradiation of AM1.5G simulated sunlight with the presence of benzyl alcohol as the hole scavenger. As shown in Supplementary Figure 16, the ZrS₁₋ᵧS₂₋ₓ(15/100) NBs with the x = 0.36 and y = 0.13 (16.3% sulfur vacancies) exhibit the best performance for photocatalytic H₂O₂ generation. When the Li amount and annealing time were simultaneously less than 150 mg and 20 min, respectively, the photocatalytic activity increased with the increase of both $S_2^{2-}$ and $S^{2-}$ vacancies. When the Li amount reached 150 mg or the annealing time reached 20 min, the photocatalytic activity showed an increase at the early stage and then decreased with the increase of annealing time or Li amount, respectively. This indicates that it is harmful to further improve photocatalytic activity with excessive either $S_2^{2-}$ or $S^{2-}$ vacancies. This is because excessive sulfur vacancies could act as the recombination centers for photogenerated charge carriers for photogenerated charge carriers. To determine the H₂O₂ formed rate, the production was analyzed by iodometry (Supplementary Fig. 17)[12]. The ZrS₁₋ᵧS₂₋ₓ(15/100) NBs possess a high H₂O₂ evolution rate of 89.6 ± 1.5 µmol h⁻¹ with good reproducibility (Supplementary Fig. 18) in the presence of benzyl alcohol as the hole scavenger (entry 5 in Supplementary Table 2), which is higher than most previous reports (Supplementary Table 3). The

wavelength-dependent apparent quantum yield (AQY) for the H₂O₂ generation on ZrS₁₋ᵧS₂₋ₓ(15/100) agrees well with its absorption spectrum, revealing that the photocatalytic activity originates from the bandgap excitation of ZrS₁₋ᵧS₂₋ₓ(15/100) (Fig. 4b). In particular, ZrS₁₋ᵧS₂₋ₓ(15/100) produces an AQY of 11.4 and 10.8% for the incident light of 400 and 500 nm respectively and demonstrates a good activity even with the excitation extended to the near-infrared region of ~700 nm. Furthermore, the photocatalyst of ZrS₁₋ᵧS₂₋ₓ(15/100) is able to maintain its activity after being recycled for the same reaction with both presence of benzylamine and benzyl alcohol, as presented in Fig. 4c and Supplementary Fig. 19a. After the stability measurement, no noticeable change was observed in the XRD patterns and Raman spectra, revealing good structure stability (Supplementary Fig. 19b and c). In addition, the S 2p XPS spectra of ZrS₁₋ᵧS₂₋ₓ(15/100) after the repeated photoreaction show a weak peak located at ~168.7 eV (Supplementary Fig. 19d), suggesting a slight surface oxidation of ZrS₁₋ᵧS₂₋ₓ(15/100) after the photocatalytic measurement.

Based on the high activity of ZrS₁₋ᵧS₂₋ₓ(15/100) for H₂O₂ generation, we further utilized benzylamine to substitute the hole scavenger. The H₂O₂ evolution rate of ZrS₁₋ᵧS₂₋ₓ(15/100) was decreased to 78.1 ± 1.5 µmol h⁻¹ with the same molar amount of benzylamine as benzyl alcohol, due to the slower oxidation kinetics of benzylamine than that of benzyl alcohol. Simultaneously, the benzylamine was oxidized and converted to benzonitrile at a rate of 32.0 ± 1.2 µmol h⁻¹ with a high selectivity of >99% (entry 2 in Supplementary Table 2 and Fig. 4d), and no other by-products were detected by the Gas Chromatography-Mass Spectrometry measurements (Supplementary Fig. 20), consistent with the previous report[14,29]. Besides, the ZrS₁₋ᵧS₂₋ₓ(15/100) photocatalyst shows the rates for decomposition of H₂O₂ of 0.14 and 0.16 h⁻¹ with the presence of benzyl

alcohol and benzylamine, respectively, and the rates for formation of $H_2O_2$ of 125 and 113 μmol $h^{-1}$ with the presence of benzyl alcohol and benzylamine, respectively (Supplementary Fig. 17b). Similar photocatalytic behaviors were also identified on both $ZrSS_{2-x}(15)$ and $ZrS_3$ NBs, which produced the $H_2O_2$ at a rate of 58.5 ± 1.7 and 30.3 ± 1.3 μmol $h^{-1}$ with the hole scavenger (entry 3 and 1 in Supplementary Table 2), respectively. As a comparison, $ZrSS_{2-x}(15)$ and $ZrS_3$ show a decreased $H_2O_2$ evolution rate of 48.0 ± 1.2 and 18.1 ± 1.2 μmol $h^{-1}$ with the use of benzylamine, and the corresponding benzonitrile generation rates are 20.7 ± 1.2 and 7.0 ± 1.0 μmol $h^{-1}$, respectively (Fig. 4d). As a result, the comparison of photocatalytic performance among $ZrS_{1-y}S_{2-x}(15/100)$, $ZrSS_{2-x}(15)$, and $ZrS_3$ reveals the key role of $S_2^{2-}$ and $S^{2-}$ vacancies on the $O_2$ reduction and benzylamine oxidation.

To provide a deep insight into the effect of defective structures in $ZrS_3$ NBs on its photocatalytic performance, the transient open-circuit potential measurements were performed on $ZrS_3$, $ZrSS_{2-x}(15)$, and $ZrS_{1-y}S_{2-x}(15/100)$ NBs to reveal the lifetime of photo-induced charge carriers (Supplementary Fig. 21 and Supplementary Equation 4)[64]. After introducing $S_2^{2-}$ vacancies, the carrier lifetime of $ZrSS_{2-x}(15)$ was significantly increased to 0.69 s as compared to 0.3 s of $ZrS_3$, while the $ZrS_{1-y}S_{2-x}(15/100)$ exhibits a further enhanced lifetime of 0.82 s, as shown in Fig. 5a. The increased photocurrent for the defective $ZrS_3$ also suggests the role of $S_2^{2-}$ and $S^{2-}$ vacancies on improving the carrier lifetime and dynamics (Supplementary Fig. 22). In order to explore the underlying mechanism for the lifetime enhancement, the charge carrier dynamics of these samples were extracted through the Mott–Schottky method. According to the Mott–Schottky equation (Supplementary Equation 1), the electron concentrations of $ZrS_3$, $ZrSS_{2-x}(15)$, and $ZrS_{1-y}S_{2-x}(15/100)$ NBs were calculated to be $4.00 \times 10^{18}$, $5.35 \times 10^{18}$, and $4.58 \times 10^{19}$ $cm^{-3}$, based on the estimated width of the depletion region ($w_d$) under the illumination of 55, 46, and 17 nm, respectively (Supplementary

Equation 2). The similar band bending between $ZrS_3$ and $ZrSS_{2-x}(15)$ suggests that the significantly enhanced carrier lifetime in $ZrSS_{2-x}(15)$ is attributed to the role of $S_2^{2-}$ vacancies in reducing electron-hole recombination rather than band bending, in agreement with the previous theoretical calculation[41]. The significantly reduced $w_d$ in $ZrS_{1-y}S_{2-x}(15/100)$ indicates a large electric field strength on the surface of $ZrS_{1-y}S_{2-x}(15/100)$, which can accelerate the extraction of photogenerated holes towards the surface and limit the internal band-to-band recombination. Moreover, the small $w_d$ in $ZrS_{1-y}S_{2-x}(15/100)$ results in a large conduction region for the free electrons compared to $ZrS_3$ and $ZrSS_{2-x}(15)$, which is beneficial for the electron transport.

On the other hand, the reaction kinetics of benzylamine oxidation on the photocatalysts were also investigated by the intensity-modulated photocurrent spectroscopy (IMPS). The typical IMPS plots and the generalized reaction schematics are shown in Supplementary Fig. 23, and the details for the calculation of rate constant of charge transfer ($k_t$) and surface recombination ($k_{rec}$) are discussed in the supporting information. Since the $k_t/(k_t + k_{rec})$ can evaluate the efficiency of charge carrier transfer between the catalyst and reactant[43], the clearly higher $k_t/(k_t + k_{rec})$ of $ZrS_{1-y}S_{2-x}(15/100)$ than those of $ZrSS_{2-x}(15)$ and $ZrS_3$ indicates that $ZrS_{1-y}S_{2-x}(15/100)$ possesses higher efficiency for benzylamine oxidation (Fig. 5b). Furthermore, $ZrSS_{2-x}(15)$ presents a slightly higher $k_t/(k_t + k_{rec})$ compared to $ZrS_3$, as derived from their similar behaviors of $k_t$ and $k_{rec}$ (Fig. 5c and d). The similar $k_{rec}$ between $ZrSS_{2-x}(15)$ and $ZrS_3$ is mainly ascribed to their similar surface band bending extracted from the Mott-Schottky results (Supplementary Fig. 7d). $ZrS_{1-y}S_{2-x}(15/100)$ with the larger surface band bending thus shows a significantly decreased $k_{rec}$. The slightly decreased $k_{rec}$ of $ZrSS_{2-x}(15)$ compared to $ZrS_3$ orginates from the suppression of surface charge recombination by $S_2^{2-}$ vacancies. The similar behavior of $k_t$ for $ZrSS_{2-x}(15)$ and $ZrS_3$ indicates that the introduction of $S_2^{2-}$

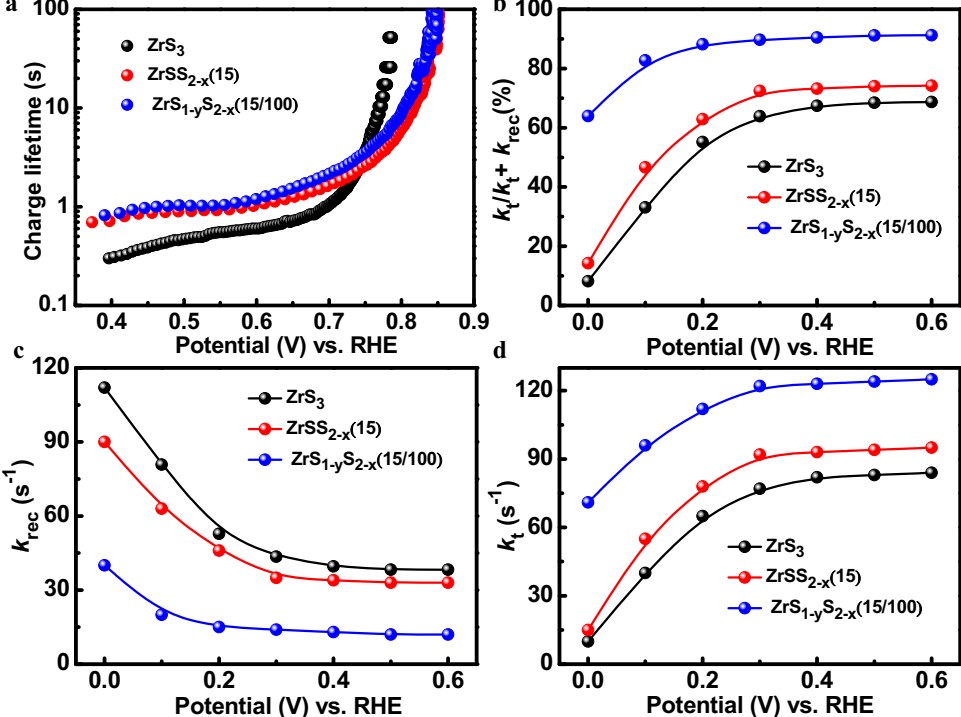

**Fig. 5 The charge carrier dynamics of the photocatalysts. a** The charge carrier lifetime of $ZrS_3$, $ZrSS_{2-x}(15)$, and $ZrS_{1-y}S_{2-x}(15/100)$ NBs. **b** Ratio of $k_t/(k_t + k_{rec})$, rate constants **c** $k_t$ and **d** $k_{rec}$ of $ZrS_3$, $ZrSS_{2-x}(15)$, and $ZrS_{1-y}S_{2-x}(15/100)$ for benzylamine oxidation measured in 0.5 M $Na_2SO_4$ with 0.1 M benzylamine.

vacancies have a subtle effect on its catalytic capability for benzylamine oxidation. Furthermore, the large increase of $k_t$ for $ZrS_{1-y}S_{2-x}(15/100)$ compared to $ZrSS_{2-x}(15)$ and $ZrS_3$ suggests that the $S^{2-}$ vacancies can act as an additional photocatalytic layer for the benzylamine oxidation (Fig. 5d).

In summary, we have developed an efficient photocatalyst of $ZrS_{1-y}S_{2-x}(15/100)$ NBs with $S_2^{2-}$ and $S^{2-}$ vacancies for the integration of photocatalytic $H_2O_2$ generation with the selective oxidation of benzylamine to benzonitrile in water. More importantly, the unique $S_2^{2-}$ vacancies and $S^{2-}$ vacancies can be controllably induced in the defective $ZrS_3$ NBs by varying the annealing time and Li amount, which promise a prospective strategy for defect engineering. With the introduction of $S_2^{2-}$ vacancies, the charge carrier recombination is prominently suppressed, and the surface $S^{2-}$ vacancies are revealed to improve the electron conduction, surface hole extraction, and kinetics of benzylamine oxidation. As a result, the photocatalyst of $ZrS_{1-y}S_{2-x}(15/100)$ exhibits a high generation rate of 78.1 ± 1.5 and 32.0 ± 1.2 μmol h$^{-1}$ for $H_2O_2$ and benzonitrile, respectively. Furthermore, $ZrS_{1-y}S_{2-x}(15/100)$ NBs possesses a photoexcitation up to ~700 nm and delivers a high AQY of 11.4 and 10.8% under the incident light of 400 and 500 nm, respectively.

## Methods

**Preparation of $ZrS_3$, $ZrSS_{2-x}$, and $ZrS_{1-y}S_{2-x}$ NBs.** The $ZrS_3$ NBs were synthesized through a typical chemical vapor transport process. 0.96 g S (99.5% purity, Alfa Aesar) and 0.91 g Zr (99.2% purity, Sigma-Aldrich) powders were mixed, and 5mg iodine (99.5% purity, Alfa Aesar) was added as a transport agent. The mixture was sealed in a quartz ampoule (Φ 6 mm × 200 mm) under the vacuum of $10^{-3}$ Pa, which was subsequently placed in the center of a two-zone furnace with a temperature gradient of ca. 15 K/cm from center to edge. The furnace was heated to 650 °C and last for 10 h to produce $ZrS_3$ powder, which has a pure monoclinic crystal structure that is stable at this temperature. The obtained 1.87 g $ZrS_3$ powder was then dispersed in isopropanol (≥99.5% purity, Alfa Aesar) at a concentration of 0.5 mg ml$^{-1}$ followed by the sonication for 15 min. The dispersion was subsequently centrifuged for 10 min at 1006 xg to remove large aggregates. Finally, about 0.6 g $ZrS_3$ NBs (32% yield) were obtained by the collection from the rest of the dispersion by further centrifugation for 10 min at 16099 xg.

Since hexagonal $ZrS_2$ (ICCD PDF no. 11-0679) is usually obtained by vacuum annealing of monoclinic $ZrS_3$ at elevated temperature (820 °C)[46]. This suggests that $ZrS_3$ can be desulfurized into $ZrS_2$ by the post-annealing at a higher temperature under vacuum. It is implied that such transformation can also be realized by the desulfuration of $S_2^{2-}$ ions, based on our previous results on $TiS_3$ and crystal structure analysis between $ZrS_2$ and $ZrS_3$ in Fig. 1 and Supplementary Fig. 1[40]. As a result, the $ZrSS_{2-x}$ NBs were prepared by the previously reported vacuum annealing method. Specifically, 0.6 g $ZrS_3$ NBs were sealed in the quartz ampule (Φ 6 mm × 10 mm) again, which was then heated to 700 °C and last for different time (10, 15, and 20 mins) to fabricate $ZrSS_{2-x}$ NBs. Besides, a certain amount of Li metal pieces (50, 100, and 150 mg) were added into 30 ml ethanediamine (≥98% purity, Sigma-Aldrich) for continuous magnetic stirring in an Ar-filled glovebox ($O_2$, $H_2O$ < 0.1 ppm). After the Li was completely dissolved, 0.5 g $ZrSS_{2-x}$ NBs were added into the solution, and the obtained solution was subsequently transferred into a 50 mL Teflon-lined autoclave and sealed immediately. Then, the Teflon-lined autoclave was taken out of the glovebox and kept in an oven at 120 °C for 24 h. After cooling down to room temperature, the mixture was first washed in 0.2 M HCl and then rinsed several times in deionized water and ethanol, where the $ZrS_{1-y}S_{2-x}$ NBs (yield > 96%) was finally obtained.

**Characterization of photocatalysts.** UV-Vis-NIR spectrometer (Hitachi U4100), field emission SEM (FE-SEM, JEOL JSM6700F), TEM (FEI Titan 80-300, operated at 200 kV), XRD (Bruker D8 Advance), XPS (ESCALAB 250Xi) with Al K$a$ X-ray as the excitation source, EPR (JEOL FA200), tapping-mode AFM (MPF-3D, Asylum Research, CA, USA), and Raman spectroscopy (Horiba Jobin Yvon Modular Raman Spectrometer) with 514 nm laser excitation were employed to characterize different properties of the defective $ZrS_3$ NBs, e.g. atomic and energy band structure. In particular, the samples for the TEM measurements were suspended in ethanol and supported onto a holey carbon film on a Cu grid.

**Coupling photocatalytic $H_2O_2$ generation with selective benzylamine oxidation over $ZrS_3$, $ZrSS_{2-x}$, $ZrS_{1-y}S_{2-x}$ NBs.** 50 mg photocatalyst was dispersed in 30 ml $H_2O$ with 1 mmol benzylamine. After the sonication for a few seconds, the mixed solution was bubbled by oxygen for 30 s. Subsequently, the solution was sealed and irradiated under an AM 1.5G simulated sunlight of 100 mW cm$^{-2}$ derived from a 300 W xenon lamp fitted with an AM 1.5 filter. At certain time intervals, the solution was filtrated by a 0.22 μm Millipore filter to remove the photocatalyst. The aqueous and organic phase products were then analyzed by the iodometry and gas chromatograph (GC) measurements, respectively.

The production of $H_2O_2$ was analyzed by the iodometry[12]. Typically, 50 μL 0.4 M potassium iodide (KI, ≥99% purity, Sigma-Aldrich) aqueous solution and 50 μL 0.1 M potassium hydrogen phthalate (≥99.5% purity, Sigma-Aldrich) aqueous solution were added to 2ml obtained aqueous phase product, which was kept for 0.5 h. The mixed solution was then detected by UV–vis spectroscopy on the basis of absorbance at 350 nm, from which the quantity of generated $H_2O_2$ was estimated. In addition, to analyze organic phase product from the benzylamine oxidation, the organic liquid was first extracted using ethyl acetate (≥99.9% purity, Sigma-Aldrich) and then detected by the GC characterization. Fihu

**Photocatalytic $H_2O_2$ generation with benzyl alcohol as hole sacrificial reagent.** 50 mg catalyst was dispersed in 30 ml $H_2O$ containing 1mmol benzyl alcohol. After sonicating for a few seconds, the mixed solution was bubbled by oxygen for few seconds. Subsequently, the solution was sealed and irradiated under an AM 1.5G simulated sunlight of 100 mW cm$^{-2}$ derived from a 300 W xenon lamp fitted with an AM 1.5 filter. The amount of $H_2O_2$ was analyzed by the iodometry. For the action spectrum analysis, the reactions were performed at 298 K under monochromated light irradiation, with the $\Phi_{AQY}$ (AQY, apparent quantum yield) determined by the following Eq. (1):

$$\phi_{AQY}(\%) = \frac{\left[ H_2O_2 \text{ formed (mol)} \right] \times 2}{\left[ \text{photon number entered into the reactor (mol)} \right]} \times 100 \qquad (1)$$

**The stability test for $ZrS_{1-y}S_{2-x}(15/100)$ NBs.** The the $ZrS_{1-y}S_{2-x}(15/100)$ NBs was recovered by centrifugation for 15 min at 16,099$g$ and used for the reaction sequence, with water replaced every 3 h during photoirradiation.

**EPR trapping measurements.** 4 mg catalyst was suspended in 500 μL $CH_3OH$ containing 50 μL DMPO (Sigma-Aldrich for ESR-spectroscopy). After the sonication, the solution was irradiated by a 300 W xenon lamp with a 420 nm filter for 3 min. The resulted solution was subjected to the analysis by using a JEOL (FA200) ESR Spectrometer.

**Photoelectrochemical measurements.** The photoelectrochemical measurements were performed in a three-electrode system with an electrochemical workstation (Zahner Zennium) under an AM 1.5G simulated sunlight of 100 mW cm$^{-2}$ (150 W, Newport 94011A LCS-100). The samples on FTO substrates were firstly prepared by a typical electrophoretic deposition method. For details, 25 ml acetone solution containing 20 mg sample and 40 mg iodine was used as the electrophoresis solution. The experimental setup consists of two pieces of FTO that serve as the anodic and cathodic electrodes, respectively. The FTO substrates were immersed in the above solution in parallel with a distance of 1cm, which were kept in the solution for 5 min at a 10 V bias under the potentiostat control. After being calcined for 2 h in a vacuum oven at 100 °C, a uniform film was firmly coated on the FTO substrates. Samples on FTO substrates were directly used as the working electrode, with a Pt wire and an Ag/AgCl (KCl saturated) electrode as counter and reference electrodes respectively. All the samples were illuminated through the sample side (front-side illumination). The photoelectrochemical performance was recorded in 0.1 M $Na_2SO_4$ electrolyte with 0.1 mM benzylamine. Mott-Schottky plots were derived from impedance-potential tests conducted at a frequency of 1 kHz in dark. IMPS spectra were recorded by the Zahner Zennium C-IMPS system.

## Data availability

All data supporting the findings in the article as well as the Supplementary Information files are available from the corresponding authors on reasonable request.

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

## Acknowledgements

The authors acknowledge the financial support from Singapore National Research Foundation under the grant of NRF2017NRF-NSFC001-007, NUS Flagship Green Energy Programme, Fundamental Research Foundation of Shenzhen (No. JCYJ20190808152607389 and No. JCYJ20170817100405375), China Postdoctoral Science Foundation (2020M672794), and Shenzhen Peacock Plan (No. KQTD2016053112042971).

## Author contributions

T.Z., H.C., L.H. and C.W. conceived and designed the experiments. T.Z., Z.Y., D.W., L.X., W.Y. and Z.Y. performed the experiments. T.Z., L.X. and H.Z. analyzed the data. T.Z, S.Y., and P.X. wrote the manuscript. All authors discussed the results and commented on the manuscript.

## Competing interests

The authors declare no competing interests.

## Additional information

**Peer review information** *Nature Communications* thanks Pau Farràs, Vladimir B. Golovko, and Lei Han for their contributions to the pee review of this work. Peer review reports are available.

