## [Peer Review File · Nature Communications]

REVIEWER COMMENTS

Reviewer #1 (Remarks to the Author):

Comments

This manuscript reported the fabrication of defective ZrS₃ nanobelts (NBs) with S²⁻ and S²⁻ vacancies through two-steps of the vacuum annealing and Li-treatment. The impacts of S²⁻ and S²⁻ vacancies on modulating the charge carrier dynamics and photocatalytic performance were also systematically investigated. The experimental results demonstrated that the S²⁻ vacancies can facilitate the separation of photogenerated charge carriers, whereas the S²⁻ vacancies can not only promote the electron conduction and hole extraction in photocatalytic process also improve the kinetics of benzylamine oxidation. Under simulated sunlight illumination, the resultant defective ZrS₃ NBs exhibited high production rate of H₂O₂ (78 μmol h⁻¹) and benzonitrile (32 μmol h⁻¹). Overall, the manuscript is well organized and written. Based on the novelty and significance of this work, acceptance of this work for Nature Communications is therefore recommended after addressing the following concerns.

- As demonstrated in the introduction, the high-temperature vacuum annealing is an effective scheme to produce S²⁻ vacancies in ZrS₃, whereas in the experimental section, the ZrS₃ NBs have been already firstly synthesized under the high-temperature vacuum annealing condition. So please provide the detailed explanations why to do the 2nd annealing at vacuum condition?
- For the Li-treatment, the description is unclear for the used Li, Li powder or Li foil? In addition, as for the use of metal Li, like lithium-battery, it is necessary to operate in the glove box from a perspective of safety, whereas there are no relevant descriptions in the manuscript. Please comment.
- The valance band maximum of the resultant defective ZrS₃ NBs was pointed out to be below the oxidation potential of benzylamine, so what is the oxidation potential of benzylamine? And what are other by-products for the oxidation of benzylamine except benzonitrile? The relevant descriptions should be stated in the discussion.
- XPS characterization was performed to confirm the vacancy type. It is better to provide such table with the detailed different vacancies contents.
- The authors only provided the stability results for H₂O₂ production, so how about benzonitrile production? Besides, for metal-sulfides-based semiconductors, the intrinsic photocorrosion generally restricts significantly their durable application due to the sulfide oxidation. So the corresponding structural and composition characterizations should be performed after stability measurement, and Zr⁴⁺ ions concentration in the solution should be also measured by ICP.
- Schematic 1h and i cannot be found, so please check and provide the detailed figures.

Reviewer #2 (Remarks to the Author):

The manuscript "Efficient Photocatalytic Hydrogen Peroxide Generation Coupled with Selective Benzylamine Oxidation over Defective ZrS₃ Nanobelts" by Zhangliu Tian at al. is definitely a high quality work. Detailed characterisation of materials studied by the authors by a wide range of complimentary techniques is very impressive, demonstrating expertise of the team and access to the cutting edge equipment. Yet, should this guarantee acceptance in Nature Communications? One would hope to find the big picture novelty and unique breakthrough at this high level of impact.

The ZrS₃ nanobelts have been studied before and their interesting optical properties have already been reported (10.1002/sml.201401376 and 10.1039/C5NR09268J). Paper "The synthesis and investigation of the reversible conversion of layered ZrS₂ and ZrS₃" New J. Chem., 2020, 44, 7583 gains insight into synthesis process parameters reporting that "At 550 °C and 650 °C, ZrS₃ nanobelts with a width from 90 to 160 nm and a thickness of about 26 ± 8 nm and 6 ± 2 nm in length were experimentally synthesized". ZrS₃ was already reported as a photocatalyst material although in the case of ASTESJ paper the authors humbly referred to their material as microribbons of ZrS₃ synthesized at 650 °C (dx.doi.org/10.25046/aj040116), concluding that there is structure-sensitivity

in ZrS₃ photocatalysis: "The rates of degradation curves were associated with the ZrS₃ samples morphology; the best result revealed for microribbons ZrS₃ synthesized at 650 °C". Do note that paper in New J. Chem. using similar conditions reports nanobelts.

The authors call their materials nanobelts, but do not comment on the thickness as the authors of New J. Chem. paper do (or have I missed this?). The authors should supply histograms of length, width and thickness. Are these truly thin "belts" or do these have square-like cross-section? Nano-suffix for the material must come for materials which have sub-micron size in at least one dimension, so not having data on thickness potentially disqualifies use of such label. Good example is in "Production of phosphorene nanoribbons" paper Nature volume 568, pages 216–220 (2019), but current study must also have histogram for thickness.

Current manuscript adds value in studying defective ZrS₃ nanobelts, albeit the authors failed to explore effect of the process parameters in the synthesis of defective materials on the types and density of defects etc., and correlation with photocatalytic activity, using only one set of reaction conditions in their study without justifying their specific choice of conditions. For example, would EPR data imply formation of more of the "S₂" type vacancies in the ZrS_{1-y}S_{2-x} or is it realistic to expect that value of g will be the same for vacancies in two different S species/environments? The authors used somewhat novel treatment with 100 mg of Li at the last stage of the defect induction in their material. It is not clear how such small quantity of highly reactive Li was measured and how its oxidation was prevented. It appears that the solvent was saturated with air and there was air in the autoclave headspace... Would even minute variations here strongly affect reproducibility of synthesis? Another aspect of claimed novelty is the high rate of production of H₂O₂ in combination with oxidation of amine to nitrile (cf. use of alcohol as a sacrificial hole scavenger, which was also studied in the current manuscript and according to Table S2 gave even higher yield of H₂O₂). Firstly, even Table S2 shows examples of materials which demonstrate higher rates of H₂O₂ production (refs 8, 12, 14 16). Secondly, the rate of production are still puny! Using 50 mg of catalyst and 1 mmol, 107.15 mg of benzylamine in 30 mL of MilliQ (industry will not use such high purity) water produced only 390 μmol of H₂O₂ after 5 hrs of irradiation. This corresponds to 13 μmol/mL or 13 mmol/L concentration, which is useless in terms of chemical industry or any realistic application. Furthermore, this H₂O₂ is in the mixture with amine and nitrile, which must be extracted with (unspecified amount) of ethyl acetate (it is also not clear if the authors took partitioning coefficients into account when performing quantification of nitrile yield). Realistic applications of H₂O₂ require efficient, ideally, on-site production of highly concentrated H₂O₂, which is known to catalytically decompose on numerous materials. Would production of highly concentrated H₂O₂ be ever feasible using reported here catalysts or would these materials actually catalyse decomposition of H₂O₂ at higher concentrations (or may be even at low concentrations observed in this study and hence actual production rates are higher, but some of the H₂O₂ decomposed)? I fail to see clever control experiments in the current version of the manuscript. Finally, Re: "Fig 3 (d) H₂O₂ and BN evolution rate by the respective photocatalysts under AM1.5G simulated sunlight irradiation. Conditions: 30 ml H₂O with 1 mmol benzylamine, 50 mg photocatalysts, 1 bar O₂" Part on photocatalytic testing in the experimental section states "the mixed solution was bubbled by oxygen for 30 s" (not sure why 30 s was chosen and how error of 1 s would affect results) but there is nothing re pressurising to 1 bar, which will boost catalysis. Industry would avoid using O₂ plant where possible as it adds costs. Duration of the experiment results of which are shown in Figure 3 (d) is not stated. But even more importantly, it looks like rate of nitrile production is only 50% of H₂O₂ production rate, which means that the reaction mixture has a significant excess of amine (1 mmol was introduced at the start) and only 100s on micromoles of nitrile were produced.

This makes nitrile production in this way completely useless since: a) the rate is even lower than that of H₂O₂ production, b) organics must be extracted using huge quantity of ethyl acetate (compared to quantity of amine and nitrile) and c) excess of amine and tiny quantity of nitrile must be separated.

Re: "Intensive research has been carried out to synthesize nitriles from primary nitriles through dehydrogenation.20-25 However, most of the reactions are conducted in organic solvents under harsh conditions, such as high-temperature, exposure to high-pressure oxygen or air, and presence of oxidants." Firstly, the authors meant "primary amines". It is a shame that the authors conveniently overlooked that there are examples of this reaction photocatalysed under mild conditions. For

example, even within the references cited by the authors one could find a paper (ref 25) reporting photocatalytic conversion of benzylamine to nitrile at 30 °C temperature in water with 90% yield of benzonitrile reached after 4 hrs of irradiation with LED light and at 30 °C temperature in toluene with 46% yield of benzonitrile after 6 hrs exposure to sunlight. And it even looks like the authors of that paper used the same concentration of amine (0.1 mmol/3 mL in ref 25 cf. 1 mmol/30 mL in the current manuscript).

Basically, all the points above make novelty of making H₂O₂ and nitrile in the way reported by the authors quite low.

The level of details provided in experimental section with respect to synthesis of materials is not acceptable even for much lower level journal.

Details of apparatus used and specific conditions used must be reported in as much detail as possible. What was typical yield at each stage etc.? See my comments on the experimental section below.

One would hope that respected group leaders insisted that their students would perform such experiments in (at the very least) duplicates. However, manuscript in its current form does not convey a message that the experiments are reproducible. It is very important to prove reproducibility of results at this level of publication.

Reproducibility of photocatalysis is also hard to see in the current form of the manuscript. How homogeneous were obtained materials? i.e. would a portion of a catalyst perform in exactly the same fashion as another portion of the catalyst from the same batch?

Would catalyst made under identical conditions (batch 1 and batch 2) have identical morphology and other properties and the same catalytic performance? All these are very important questions which must be answered if the paper is to be accepted for Nature Communications. Also, having done all these experiments the authors will be able to provide values of experimental uncertainties.

Re Fig 3(c): Experimental section does NOT present details on how this recycling study was performed. Specifically, it could potentially be rather difficult to ensure that all 50 mg of the catalyst could be fully recovered at the end of one catalytic test and fully re-introduced into the next catalytic test. Is this done with benzyl alcohol or benzylamine? 225 micromoles to be divided by 3 hrs and then multiplied by 5 hrs one would get 375 micromoles, which implies that this experiment is likely to have been done with benzylamine. Yet, this also shows that uncertainties and reproducibility across wider kinetic profile (cf. 2 time points of 3hr and 5 hr) must be established.

My comments and edits of synthesis section are highlighted here: "Preparation of ZrS₃, ZrSS_{2-x} and ZrS_{1-y}S_{2-x} NBs: The ZrS₃ NBs were synthesized through a typical chemical vapor transport process. S (99.5 % purity, Alfa Aesar) and Zr (99.2 % purity, Sigma-Aldrich) powders were mixed according to a molar ratio of 1:3 (amounts used?), and 5 mg iodine (99.5 % purity, Alfa Aesar) was added as a transport agent. The mixture was sealed in a quartz ampoule (Φ 6 mm × 200 mm) under the vacuum of 10⁻³ Pa (if this is done under vacuum, why vacancies are not formed as at the next stage? Why authors did not comment on this?), which was subsequently placed in the centre of a two-zone furnace with a temperature gradient of ca. 15 K/cm from center to edge. The furnace was heated to 650 °C and last for 10 h to produce ZrS₃ powder. The obtained ZrS₃ powder was then dispersed in isopropanol (≥99.5 % purity, Alfa Aesar) at a concentration of 0.5 mg mL⁻¹ followed by the sonication for 15 min. The dispersion was subsequently centrifuged for 10 min at 3000 rpm to remove large aggregates. Finally, the ZrS₃ NBs were obtained by the collection from the rest of dispersion (yield?). The ZrSS_{2-x} NBs were prepared using the previously reported vacuum annealing method.³⁴ Specifically, the as-grown ZrS₃ NBs (typical amount used?) were sealed in the quartz ampoule again (vacuum annealing method implies use of vacuum which is clearly NOT mentioned here), which was then heated to 700 °C and maintained at that temperature for 15 mins to fabricate ZrSS_{2-x} NBs (above ZrS₃ powder was purified from large species via dispersion and centrifugation, here treatment is at even higher T, albeit duration is shorter, but there is no certainty that fusion/sintering of the NBs did not occur or the material was purified). In addition, 0.5 g ZrSS_{2-x} NBs were placed in a 50 mL Teflon-lined autoclave filled with 30 mL ethanediamine (≥98 % purity, Sigma-Aldrich) and 100 mg Li (how such small quantity of Li was measured and how its oxidation was prevented? Would minute variations here strongly affect reproducibility of synthesis?), and the autoclave was subsequently kept in an oven at 120 °C for 24 h. After cooling down to the room temperature, the mixture was first

washed in 0.2 M HCl and then rinsed several times in deionized water and ethanol, where the ZrS_{1-y}S_{2-x} NBs was finally obtained (yield?)."

In the light of the above and taking into account that there is only one photocatalytic study using similar materials and significant novelty of defect engineering in these materials, and excellent characterisation presented in the current study, I would recommend accepting this manuscript AFTER MAJOR REVISION.

The authors must tidy up experimental, proving reproducibility and reporting uncertainties, report kinetic profiles, do careful control experiments AND, IMPORTANTLY, diligently explore effects of the process parameters in the defect generating reactions on the nature/density of defects and photocatalytic performance of obtained materials. Ability to control defect engineering in these materials and through that control the catalytic activity would make this paper worthy of publication in Nature Communications. Specific photocatalytic application in this case could be of lower novelty/importance with focus on control of defect engineering.

Reviewer #3 (Remarks to the Author):

The paper "Efficient Photocatalytic Hydrogen Peroxide Generation Coupled with Selective Benzylamine Oxidation over Defective ZrS₃ Nanobelts" describes an elegant modification of ZrS₃ by introducing two types of vacancies, S₂⁻ and (S₂)₂⁻. The addition of these vacancies enhance the photoinduced charge separation with a concomitant increase in photocatalytic activity.

The authors have provided a comprehensive analysis of the materials, ZrS₃, ZrSS_{2-x} [(S₂)₂⁻ vacancies], ZrS_{1-y}S [(S₂⁻ vacancies] and ZrS_{1-y}S_{2-x} [both types of vacancies]. Characterisation through XRD, XPS, Raman, electron microscopies, electrochemistry, EPR, DRUV-vis, etc. show clear characteristics of each type of vacancy and its effect on the overall performance. Finally, the materials have been tested for the selective oxidation of benzylamine and reduction of oxygen to hydrogen peroxide, showing the highest reported yield for H₂O₂ production.

From a synthetic viewpoint, the introduction of (S₂)₂⁻ vacancies using thermal annealing has been previously described for ZrS₃, while the use of lithium to induce S₂⁻ has been reported for other materials, but not for ZrS₃. The consequence on such vacancies is clearly demonstrated and this methodology can potentially be used for other transition metal trichalcogenides. However, it is difficult to envisage such method for other type of materials.

Overall, the paper is interesting and provides proof that the presence of both vacancies enhances the performance of the photocatalysis. However, there are a number of points for consideration:

- 1) In page 4: it is introduced the use of lithium to induce vacancies, however the sentence is out of context and further explanation should be given. Are there any references of the effect of S₂⁻ vacancies on other materials?
- 2) XRD analysis: the authors mention that there is no phase transition, it is indeed true that no new peaks appear. However, the intensity of some peaks is quite different, the authors should provide an explanation. Have they performed Rietveld analysis to check changes in cell unit dimensions?
- 3) The size of the NBs is quite different, is there any effect on the catalysis the different width? After treatment to generate vacancies, are the TEM showing the same morphology? No images have been added in the SI.
- 4) XPS data: the authors mention a higher binding energy for modified ZrS₃ compared to the initial ZrS₃ NBs, however the values change from 184 eV to 183.5 eV for Zr 3d, and same shift is observed for S 2p spectra. Should read to lower binding energy. Why S₂⁻ vacancies cause a shift in all 3 peaks in S 2p, whereas (S₂)₂⁻ only change intensity? This is not clear in the manuscript.
- 5) Photocatalytic activity: there are no plots on the results obtained for any of the measurements. The authors mention that H₂O₂ is measured using the iodometry method, spectra of the different results should be added as SI. Likewise, GC is used for the oxidation of benzylamine, spectra should be added. Moreover, there is no mention of the product of benzylamine oxidation, which one is it? Which is the ratio between photocatalyst and substrate? Is it really in catalytic amounts?
- 6) Photoelectrochemical measurements: in the methods section it is mentioned that samples were coated directly on FTO, but it is unlikely that they will be stable on the surface of the electrode for the

duration of the measurements. Have the authors used any binder to fix the samples on the FTO?

7) Results from EOC decay and IMPS are not clear. Figure 4a shows a similar behaviour for ZrSS_{2-x} and ZrS_{1-y}S_{2-x}, however the data analysis to obtain the kinetic constants show a similar behaviour for ZrS₃ and ZrSS_{2-x}. Can the authors explain the reason?

8) There are a number of references that should be added:

For H₂O₂ production: <https://doi.org/10.1039/C9EE02247C>

For solar chemicals production: <https://doi.org/10.1039/C8CC02487A>

For open-circuit photocurrent: <https://doi.org/10.1021/jz300293q>

REVIEWER COMMENTS

Reviewer #1 (Remarks to the Author):

The authors have thoroughly addressed my comments and also those of the other reviewers. I believe that the quality of the revised manuscript is very good and could be accepted without change for publication.

Reviewer #2 (Remarks to the Author):

I would like to take this opportunity to congratulate the authors of this manuscript on significantly improving quality of the presented work. The only two suggestions for further improvement I would have is for the authors to show experimental uncertainties with respect to catalytic test results where possible and possibly shift the focus to the highlight of the ability to control defects in these novel materials (which in turn affect catalytic activity) as opposed to claims of the high H₂O₂ production rates etc.

I now believe that this manuscript reached novelty level and approached standards of presentation required for publication in the high impact journal, such as Nature Communications.

Reviewer #3 (Remarks to the Author):

The authors have provided a comprehensive explanation and further measurements after the previous reviewer's comments. The quality of the manuscript has improved considerably with a polished and more detailed experimental section, as well as further discussion of results. I would recommend acceptance of the manuscript after some minor corrections:

- Figure 3c: there is a typo in the y-axis
- Photoelectrochemical measurements: it is still unclear how this were done, and a detailed analysis has been performed out the data obtained from the Mott-Schottky plots. The photoelectrodes are prepared by electrophoretic deposition, however the authors use iodine as one of the reagents. Given the relevance of the results, further analysis on the photoelectrode used to analyse the samples should be provided. Is there any iodine trapped within the thin film? has the sample changed after a 10V bias? I believe it would be good to know for sure that what the authors are measuring with the photoelectrochemical set-up is actually the same sample (photocatalyst).
- For the photocatalytic recycling tests, the authors mention that they use centrifugation to separate the powder photocatalyst. Have they tried to weight the powder to see if they are able to recover 100% the photocatalyst?
- Finally, the conclusions should be expanded slightly to summarise a bit more in detail the effect of defect engineering so that it can be extrapolated to other types of materials. The last sentence " Our results promise a novel strategy for the artificial photosynthesis of liquid solar fuels and other valuable chemicals" is out of context and should be removed.

Responses to Reviewers

Reviewer 1:

Comment: The authors have thoroughly addressed my comments and also those of the other reviewers. I believe that the quality of the revised manuscript is very good and could be accepted without change for publication.

Response: We thank the reviewer for the positive comments.

Reviewer 2:

Comment: I would like to take this opportunity to congratulate the authors of this manuscript on significantly improving quality of the presented work. The only two suggestions for further improvement I would have is for the authors to show experimental uncertainties with respect to catalytic test results where possible and possibly shift the focus to the highlight of the ability to control defects in these novel materials (which in turn affect catalytic activity) as opposed to claims of the high H₂O₂ production rates etc. I now believe that this manuscript reached novelty level and approached standards of presentation required for publication in the high impact journal, such as Nature Communications.

Response: We thank the reviewer for the constructive suggestions.

All the experimental uncertainties with respect to catalytic test results have been added, as shown in **Table R1** and **Figure R1**. We have revised the manuscript and supporting information accordingly.

To shift the focus to the highlight of the ability to control defects, the detailed summary of controllable defect engineering has been added into the Abstract and Conclusion Sections.

Table R1. The activity of photocatalytic H₂O₂ generation and/or benzylamine oxidative coupling reaction.^a

Entry	Catalyst	BN rate ($\mu\text{mol h}^{-1}$)	Sel (%) ^c	H ₂ O ₂ rate ($\mu\text{mol h}^{-1}$)
1 ^b	ZrS ₃	-	-	30.3 ± 1.3
2	ZrS ₃	7.0 ± 1.0	>99%	18.1 ± 1.2
3 ^b	ZrSS _{2-x} (15)	-	-	58.5 ± 1.7
4	ZrSS _{2-x} (15)	20.7 ± 1.2	>99%	48.0 ± 1.2
5 ^b	ZrS _{1-y} S _{2-x} (15/100)	-	-	89.6 ± 1.5
6	ZrS _{1-y} S _{2-x} (15/100)	32.0 ± 1.2	>99%	78.1 ± 1.5

^a Reaction conditions: 30 ml aqueous solution with 1 mmol benzylamine, 50 mg catalysts, 1 atm O₂, AM1.5G simulated sunlight (1 sun) irradiation. ^b 1 mmol benzyl alcohol instead of benzylamine. ^c Determined by GC analysis.

Figure R1. H₂O₂ and benzonitrile evolution rate by the respective photocatalysts under AM1.5G simulated sunlight irradiation. Conditions: 30 ml H₂O with 1 mmol benzylamine, 50 mg photocatalysts, 1 bar O₂.

Changes to the manuscript: Supplementary Table 2 and Figure 3d have been

replaced by **Table R1** and **Figure R1** by adding experimental uncertainties, respectively.

The following sentence has been added to the Abstract Section: “More importantly, the S_2^{2-} and S^{2-} vacancies can be separately introduced into ZrS_3 nanobelts in a controlled manner.”

The following sentence has been added to the Conclusion Section: “More importantly, the unique S_2^{2-} vacancies and S^{2-} vacancies can be controllably induced in the defective ZrS_3 NBs by varying the annealing time and Li amount, which promise a novel strategy for defect engineering.”

Reviewer 3:

Comment 1: The authors have provided a comprehensive explanation and further measurements after the previous reviewer's comments. The quality of the manuscript has improved considerably with a polished and more detailed experimental section, as well as further discussion of results. I would recommend acceptance of the manuscript after some minor corrections:

Response: We thank the reviewer for the positive comments.

Comment 2: Figure 3c: there is a typo in the y-axis.

Response: Figure 3c has been corrected, which is shown in **Figure R2**.

Figure R2. Results of H₂O₂ and benzonitrile generation for a repeated photoreaction sequence with ZrS_{1-y}S_{2-x}(15/100) under AM1.5G simulated sunlight irradiation.

Comment 3: Photoelectrochemical measurements: it is still unclear how this were done, and a detailed analysis has been performed out the data obtained from the Mott-Schottky plots. The photoelectrodes are prepared by electrophoretic deposition, however the authors use iodine as one of the reagents. Given the relevance of the results, further analysis on the photoelectrode used to analyse the samples should be provided. Is there any iodine trapped within the thin film? has the sample changed after a 10V bias? I believe it would be good to know for sure that what the authors are measuring with the photoelectrochemical set-up is actually the same sample (photocatalyst).

Response: We thank the reviewer for the constructive comment. The photoelectrochemical measurements were performed in a three-electrode system with an electrochemical workstation, and the Mott-Schottky plots were measured by using this system in 0.5 M Na₂SO₄ with 0.1 M benzylamine without illumination. The experimental details have been described in the Method Section: “*The photoelectrochemical measurements were performed in a three-electrode system with an electrochemical workstation (Zahner Zennium) under an AM 1.5G simulated*

sunlight of 100 mW cm⁻² (150 W, Newport 94011A LCS-100). Samples on FTO substrates were directly used as the working electrode, with a Pt wire and an Ag/AgCl (KCl saturated) electrode as counter and reference electrodes respectively. All the samples were illuminated through the sample side (front-side illumination). The photoelectrochemical performance was recorded in 0.1 M Na₂SO₄ electrolyte with 0.1 mM benzylamine. Mott-Schottky plots were derived from impedance-potential tests conducted at a frequency of 1 kHz in dark. IMPS spectra were recorded by the Zahner Zennium C-IMPS system.”

To check whether any iodine is trapped within the thin film, the SEM-EDS measurements were conducted on the sample of ZrS_{1-y}S_{2-x}(15/100) deposited on the FTO substrate. The SEM-EDS mapping image (**Figure R3a**) and the corresponding EDS spectrum (**Figure R3b**) clearly reveal that no iodine is trapped within the film. The sample on the FTO substrate was further analyzed by Raman measurement. As shown in **Figure R4**, no obvious change was observed in the Raman spectra after the deposition of ZrS_{1-y}S_{2-x}(15/100) on the FTO substrate, indicating that the deposition process does not affect the photocatalyst.

Figure R3. (a) SEM-EDS mapping image of the $ZrS_{1-y}S_{2-x}(15/100)$ NBs deposited on the FTO substrate and (b) the corresponding EDS spectrum.

Figure R4. Raman spectra of the $\text{ZrS}_{1-y}\text{S}_{2-x}(15/100)$ NBs before and after the deposition on the FTO substrate.

Changes to the manuscript: Figure R3a, b, and R4 have been added to the Supporting Information as **Supplementary Figure 6a, b, and c**. The following sentences have been added to the last paragraph on page 7: “The Mott–Schottky plots for all three samples exhibit positive slopes, indicating the n-type behavior of ZrS_3 (Figure 1f). **These results were obtained by measuring the photocatalysts deposited on the fluorine-doped tin oxide (FTO) substrate. It is worth noting that the deposition process did not induce any obvious change of the photocatalyst (Supplementary Figure 6a-c), suggesting that the sample on the FTO substrate measured with the photoelectrochemical set-up is essentially the same photocatalyst.**”

Comment 4: For the photocatalytic recycling tests, the authors mention that they use centrifugation to separate the powder photocatalyst. Have they tried to weight the powder to see if they are able to recover 100% the photocatalyst?

Response: We have weighed the photocatalyst after recovered by centrifugation and a recovery rate of >99 % can be achieved.

Comment 5: Finally, the conclusions should be expanded slightly to summarise a bit more in detail the effect of defect engineering so that it can be extrapolated to other

types of materials. The last sentence "Our results promise a novel strategy for the artificial photosynthesis of liquid solar fuels and other valuable chemicals" is out of context and should be removed.

Response: We thank the reviewer for the constructive comment. The manuscript has been revised accordingly.

Changes to the manuscript: The following sentence has been added to the Abstract Section: “More importantly, the S_2^{2-} and S^{2-} vacancies can be separately introduced into ZrS_3 nanobelts in a controlled manner.”

The following sentence has been added to the Conclusion Section: “More importantly, the unique S_2^{2-} vacancies and S^{2-} vacancies can be controllably induced in the defective ZrS_3 NBs by varying the annealing time and Li amount, which promise a novel strategy for defect engineering.”

REVIEWERS' COMMENTS

Reviewer #3 (Remarks to the Author):

The authors have properly addressed my comments and those from the other reviewers. I recommend publication of the manuscript without any further modifications.